# Structured Partial Stochasticity in Bayesian Neural Networks

**Tommy Rochussen**                                            TNR22@CANTAB.AC.UK

## Abstract

Bayesian neural network posterior distributions have a great number of modes that correspond to the same network function. The abundance of such modes can make it difficult for approximate inference methods to do their job. Recent work has demonstrated the benefits of partial stochasticity for approximate inference in Bayesian neural networks; inference can be less costly and performance can sometimes be improved. I propose a structured way to select the deterministic subset of weights that removes neuron permutation symmetries, and therefore the corresponding redundant posterior modes. With a drastically simplified posterior distribution, the performance of existing approximate inference schemes is found to be greatly improved.

## 1. Introduction

While neural networks boast phenomenal utility across a broad range of applications, they are often avoided in settings for which uncertainty estimation is key. In theory, the Bayesian formalism empowers neural networks to produce accurate uncertainty estimates, but unfortunately existing approximate inference schemes are too limited in either accuracy or efficiency to be practical in such scenarios (Papamarkou et al., 2024). It is then of critical importance to find better ways to approximate Bayesian inference in neural networks.

Many neural network architectures contain symmetries in their weights such that networks with different weights settings can be functionally equivalent (Chen et al., 1993). A common example is that of same-layer neuron permutations in feed-forward architectures; if two neurons in the same layer of a neural network are swapped, then as long as the weight matrices either side of the layer are permuted accordingly, the network is functionally identical. This poses an unidentifiability problem when trying to estimate network weights from data, and, because of the symmetry of the isotropic priors that are generally used, causes Bayesian neural network (BNN) posteriors to have factorially many (in the number of neurons) identically shaped modes (Ainsworth et al., 2023). In the *maximum a posteriori* (MAP) setting, this is not such a problem since gradient ascent algorithms can drive the parameters to the maximum of any such functionally equivalent modes, but it creates a significant challenge for approximate inference schemes. It has been conjectured that BNN posteriors could be quite simple—potentially even unimodal—if such parameter symmetries were accounted for (Entezari et al., 2022; Rossi et al., 2023), and so the challenge of performing approximate inference in BNNs would be made easier if such symmetries could be removed.

In general, the symmetry removal problem can be approached by altering either the prior or the likelihood, but since prior-altering approaches are "soft" in the sense that they only *suppress* redundant modes rather than altogether *remove* them (Kurle et al., 2021,

2022), I pursue the latter approach in this work. In particular, I propose to fix certain weights in the network as constants so as to purposefully break the intra-layer permutation symmetry, removing all but one of the equivalent modes and yielding improved performance from existing approximate inference schemes.

## 2. Background

### 2.1. Neuron Permutation Symmetries

In this work, I focus on neuron permutation symmetries in multilayer perceptrons (MLPs), but the procedures developed can be adapted for other feed-forward architectures.

Let $f_{\mathcal{W}}(\cdot)$ denote an $L$-layered MLP parameterised by weights $\mathcal{W} = \{\mathbf{W}_\ell\}_{\ell=1}^L$ and biases $\mathcal{B} = \{\mathbf{b}_\ell\}_{\ell=1}^L$ such that $\mathbf{W}_\ell \in \mathbb{R}^{D_{\ell-1} \times D_\ell}$ and $\mathbf{b}_\ell \in \mathbb{R}^{D_\ell}$. A forward pass of activation $\mathbf{h}_{\ell-1} \in \mathbb{R}^{D_{\ell-1}}$ through layer $\ell$ is computed as

$$\mathbf{h}_\ell = \phi_\ell\left(\mathbf{h}_{\ell-1}\mathbf{W}_\ell + \mathbf{b}_\ell\right)$$

where $\phi_\ell(\cdot)$ denotes the layer's elementwise activation function. The permutation symmetry can be observed by considering the following application of a permutation matrix $\mathbf{P} \in \mathbb{R}^{D_\ell \times D_\ell}$ and its inverse $\mathbf{P}^T$:

$$\begin{aligned}
\mathbf{h}_{\ell+1} &= \phi_{\ell+1}\Big(\left(\mathbf{h}_\ell\mathbf{P}^T\mathbf{P}\right)\mathbf{W}_{\ell+1} + \mathbf{b}_{\ell+1}\Big) \\
&= \phi_{\ell+1}\Big(\phi_\ell\big(\mathbf{h}_{\ell-1}\mathbf{W}_\ell + \mathbf{b}_\ell\big)\mathbf{P}^T\mathbf{P}\mathbf{W}_{\ell+1} + \mathbf{b}_{\ell+1}\Big) \\
&= \phi_{\ell+1}\Big(\phi_\ell\big(\mathbf{h}_{\ell-1}\underbrace{\mathbf{W}_\ell\mathbf{P}^T}_{\mathbf{W}_\ell'} + \underbrace{\mathbf{b}_\ell\mathbf{P}^T}_{\mathbf{b}_\ell'}\big)\underbrace{\mathbf{P}\mathbf{W}_{\ell+1}}_{\mathbf{W}_{\ell+1}'} + \mathbf{b}_{\ell+1}\Big)
\end{aligned} \tag{1}$$

where the third line holds due to the elementwise nature of the activation functions. Section 4.1 summarises existing work on parameter symmetry removal for improved approximate inference.

## 3. Symmetry Breaking via Structured Partial Stochasticity

### 3.1. Structured Light Pruning

One of the most straightforward ways to break symmetry in an architecture is to remove specific connections between neurons in adjacent layers; to *prune* specific weights. Consider removing the connection between the top neuron of layer $\ell - 1$ and the top neuron of layer $\ell$, the second top neuron of layer $\ell - 1$ and the second top neuron of layer $\ell$, and so on for all of the $\min(D_{\ell-1} - 1, D_\ell - 1)$ top neurons of layers $\ell - 1$ and $\ell$. If we carry out the same procedure, but this time for the *bottom* $\min(D_\ell - 1, D_{\ell+1} - 1)$ neurons of layers $\ell$ and $\ell+1$, then the number of neurons in layer $\ell$ that are fully connected to both adjacent layers, $D_\ell^{(fc)}$, is given by[1]:

$$D_\ell^{(fc)} = \max\big(0, D_\ell - D_{\ell-1} - D_{\ell+1} + 2\big) \tag{2}$$

---

1. Note that if $D_\ell - D_{\ell-1} - D_{\ell+1} + 2$ is negative, then its magnitude represents the number of neurons in layer $\ell$ that have a pruned connection on *both* sides.

and every other neuron in the layer will be connected to neurons in the adjacent layers in a unique way. Then, so long as $D_\ell^{(fc)} \leq 1$, we will have removed all of the permutation symmetries from layer $\ell$ since every neuron in the layer will have a unique set of connections. Note that this is equivalent to applying a mask to the weights $\mathbf{W}_i$ with zeroes on the first or last $\min(D_{i-1} - 1, D_i - 1)$ elements of the leading or trailing diagonal respectively for $i \in \{\ell, \ell + 1\}$. To construct a network without permutation symmetries, the procedure is carried out from the second to the penultimate layer, and connections are removed between the neurons at the tops of adjacent layers and bottoms of adjacent layers in alternating fashion—but each layer must satisfy the above inequality. Of the two schemes I present, this will be referred to as the *light* scheme due to the relatively small number of connections that are removed.

## 3.2. Structured Heavy Pruning

While the above scheme is sufficient to remove the permutation symmetries in theory, if certain non-pruned weights are close to zero in value then some permutation symmetries may remain mostly intact. To mitigate this, I propose the following scheme instead. To remove the permutation symmetries from layer $\ell$, remove the connection between the top neuron of layer $\ell - 1$ and the top neuron of layer $\ell$, the connections between the second top neuron of layer $\ell - 1$ and the *top two* neurons of layer $\ell$, the third top neuron of layer $\ell - 1$ and the *top three* neurons of layer $\ell$, and so on for all of the $\min(D_{\ell-1}-1, D_\ell-1)$ top neurons of layers $\ell - 1$ and $\ell$. As before, repeat the procedure for the bottom $\min(D_\ell - 1, D_{\ell+1} - 1)$ neurons of layers $\ell$ and $\ell + 1$. Equation (2) is still used to compute $D_\ell^{(fc)}$ for this scheme, and $D_\ell^{(fc)} \leq 1$ must still be true for there to be no symmetries. Similarly, this scheme is equivalent to applying a mask where, for the first (or last) $\min(D_{i-1} - 1, D_i - 1)$ rows, the elements on the leading (or trailing) diagonal and below (or above) are zeroes. The advantage of this scheme, which will be referred to as the *heavy* pruning scheme, is that many more weights are set to zero and therefore many more non-pruned weights would have to be close to zero for a permutation symmetry to persist. Furthermore, since the output scale of the network is fixed, pruning more weights will require the non-pruned weights to take larger values in general, forcing them away from zero.

## 3.3. Partial Stochasticity as a Generalisation of Pruning

Although increasing the number of pruned connections is likely to remove permutation symmetries more effectively, there is an alternative direction through which we can make it harder for symmetries to remain. If the priors that we commonly use truly reflect our beliefs, then we expect zero to be the modal weight value and therefore that it should be quite common for symmetries to remain as described in Section 3.2. However, since "removing a connection" just means fixing a weight's value to zero, we can generalise this to *nonzero* values instead—values that are less probable under the prior and therefore less likely to permit any permutation symmetries to remain. I propose two methods for choosing fixed nonzero values:

1. select a constant $c$ and permanently fix the value of the relevant weights to either $c$ or $-c$ with equal probability,

2. obtain a set of MAP weights and permanently fix the value of the relevant weights to their corresponding MAP counterpart.

The weights that are fixed in this way are then treated as deterministic, and (approximate) inference can be performed over the remaining weights with the great advantage that the posterior distribution of interest should contain no repeated modes caused by permutation symmetries.

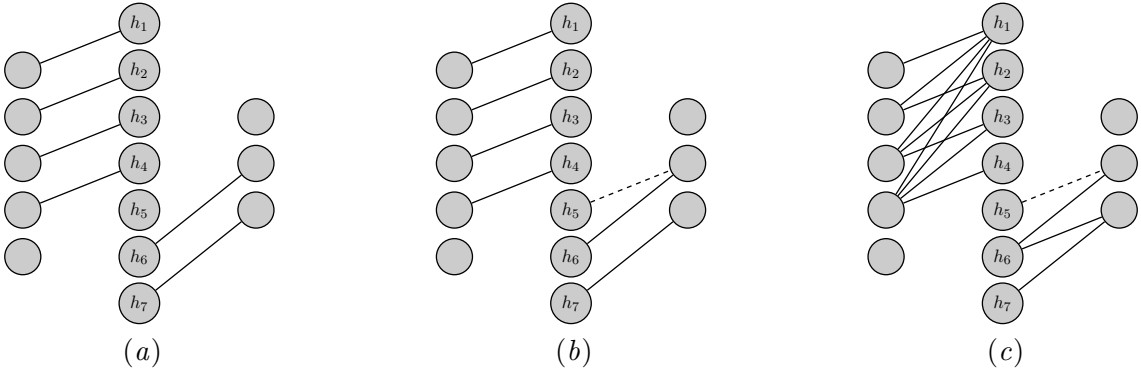

$(a)$ $\qquad$ $(b)$ $\qquad$ $(c)$

Figure 1: Connections that are fixed under structured partial stochasticity[3]. Figure 1$(a)$ and Figure 1$(b)$ correspond to the light scheme, while Figure 1$(c)$ corresponds to the heavy scheme. Connections with weights that are free are omitted except for a connection (dashed line) from neuron $h_5$ in Figure 1$(b)$ and Figure 1$(c)$, whose weight takes near-zero value.

Figure 1 depicts the inter-neuronal connections corresponding to fixed weights under the proposed schemes for a single-hidden-layer architecture. For a permutation symmetry to exist in the hidden layer, there must be at least one pair of neurons in the hidden layer that can be swapped without altering the structure of the fixed connections. Although this is not possible in Figure 1$(a)$, we see that if the value of a weight, such as the one denoted by the dashed line in Figure 1$(b)$, is very close to zero under the posterior then a swap between neurons $h_5$ and $h_6$ can be performed with (close to) no change to the overall network, assuming pruning is the fixing policy. However, Figure 1$(c)$ demonstrates that such a swap is no longer possible under the heavy scheme of partial stochasticity. Biases are omitted here since their inclusion would change nothing other than to clutter the diagrams. See Appendix A for a linear-algebraic view of the proposed schemes.

---

3. If pruning is the fixing scheme, the drawn connections can be viewed as "holes" in a similar way to missing electrons in the field of semiconductors.

## 4. Related Work

### 4.1. Parameter Symmetries in Bayesian Neural Networks

The effects of parameter symmetries on BNN posteriors has garnered an increasing amount of interest in recent years. Pourzanjani et al. (2017) introduce a bias-ordering constraint to remove permutation symmetries and a unit-length weight vector constraint to remove scaling symmetries introduced by piece-wise linear activation functions such as the ReLU. Kurle et al. (2021) argue that if biases are near zero then permutation symmetries can remain mostly intact under such a constraint, and so instead they propose the use of skip connections with fixed matrices. My approach is somewhat similar to this, but I leave the model architecture unchanged and instead fix a subset of the weights. Seperately, Kurle et al. (2022) investigate the impact of parameter symmetries on mean-field variational inference (MFVI) (Blundell et al., 2015) in particular, and they find that some of the known pathologies of MFVI in BNNs—such as underfitting (Dusenberry et al., 2020) or collapsing to the prior (Coker et al., 2022)—can be attributed to the inability of MFVI to model the repeated modes in the posterior, and that the pathology could be eliminated by accounting for symmetries in the variational objective.

Rossi et al. (2023) perform a similar study to Ainsworth et al. (2023) but in the context of approximate Bayesian posteriors instead of MAP point estimates. They show empirically that if two independently obtained variational posteriors for a network are aligned with respect to weight permutations, then the parameters of the two variational posteriors can be linearly interpolated without degradation to the posterior approximation. Similarly, Xiao et al. (2023) use the mode-alignment algorithm developed by Ainsworth et al. to map samples obtained from ensembles (Lakshminarayanan et al., 2017) or separate runs of Hamiltonian Monte Carlo (HMC) (Neal, 1992) to the same mode, before fitting a diagonal Gaussian to the aligned samples which can be sampled from cheaply. Wiese et al. (2023) show theoretically that sampling from a small selection of posterior modes is enough to target the exact posterior predictive distribution due to parameter symmetries, and so they, as well as Sommer et al. (2024), propose variations of using samples from separate HMC chains as opposed to one longer chain.

### 4.2. Partial Stochasticity in Bayesian Neural Networks

Performing inference over a subset of a neural network's parameters is often seen as a way to trade approximate inference accuracy for scalability. Daxberger et al. (2021) are motivated to limit the stochasticity so that costlier high fidelity approximate inference schemes can be used on the stochastic subset, but their goal remains to approximate the vanilla BNN posterior. Similarly, neural linear models (Ober and Rasmussen, 2019) allow for tractable Bayesian inference but are *not* universal conditional distribution approximators (Sharma et al., 2023). Furthermore, Kristiadi et al. (2020) leverage partial stochasticity as a way of improving the uncertainty calibration of MAP networks, but not as a way to improve approximate inference.

Conversely, Sharma et al. (2023) challenge the notion that partial stochasticity cannot improve upon full stochasticity. They show that certain architectures of partially stochastic network are universal conditional distribution approximators, that high and low fidelity

approximate inference schemes in partially stochastic networks can rival or even supersede the same scheme in fully stochastic ones, and that significant computational savings may be enjoyed all the while. I believe that part of the success of Sharma et al.'s partial stochasticity can be attributed to the fact that they inadvertently remove parameter symmetries. One of the schemes they propose involves fixing entire layers to their MAP setting; if $\mathbf{W}_\ell$ is stochastic but $\mathbf{W}_{\ell+1}$ is fixed, then Equation (1) demonstrates that no permutation can be applied to $\mathbf{h}_\ell$ without also permuting the fixed weights, and therefore many of $\mathbf{W}_\ell$'s permutation symmetries will have been removed. The difference between their work and this is that here the choice of which parameters to fix is based on a symmetry-breaking structure in the hope of removing *all* permutation symmetries.

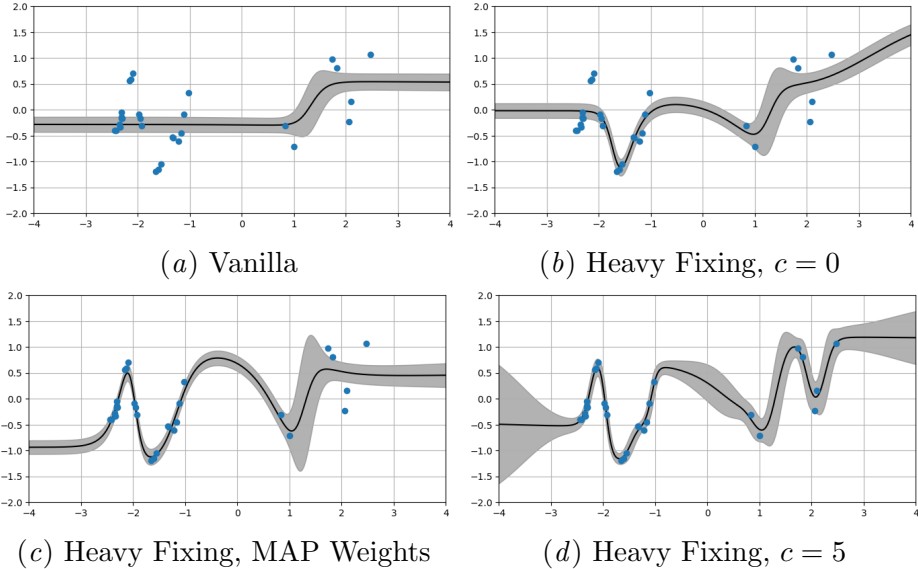

(*a*) Vanilla

(*b*) Heavy Fixing, $c = 0$

(*c*) Heavy Fixing, MAP Weights

(*d*) Heavy Fixing, $c = 5$

Figure 2: One-dimensional regression. Blue circles indicate datapoints, black lines and shaded areas represent predictive means and 95% confidence regions respectively.

## 5. Experiments

### 5.1. Toy Regression

I evaluate my method on a 1D artificial regression dataset, using networks with two hidden layers of 50 units and MFVI as the approximate inference method. The likelihood variance was set to that of the data-generating process. The results are found in Figure 2. Compared to vanilla MFVI, there is a clear improvement in both predictive mean and uncertainty across all methods, but particularly in the heavy fixing with $c = 5$ case which displays impressive in-between uncertainty. Although a smaller likelihood variance reduces the underfitting of vanilla MFVI (but increases overconfidence), I would argue that an approximate inference scheme should be capable of good performance under the "correct" noise model. See Appendix B.1 for further experimental details.

## 5.2. Energy Efficiency

I evaluate all proposed schemes on the UCI Energy Efficiency regression dataset (Tsanas and Xifara, 2012). Using networks with two hidden layers of 50 units under MFVI once more, I compare my approach to a vanilla MFVI scheme, as well as two schemes in which no symmetry-breaking structure is used, but rather a random subset of weights are randomly set to $\pm c$. The size of the random subsets are the same as in the light and heavy schemes respectively. $c$ was optimised jointly with the variational parameters and the likelihood variance was set to the value that maximised the test-set accuracy of a MAP network. I report the test-set root-mean squared error (RMSE) and log posterior predictive (LPP) probability as accuracy and calibration metrics respectively. The results are found in Table 1. All six schemes result in improved MFVI performance, and the relatively poor performance of the two schemes with no symmetry-breaking structure demonstrates that it is indeed the symmetry-breaking that is responsible for improved approximate inference. See Appendix B.2 for further experimental details, including errorbars of results.

|              | HF        | LF    | HP    | LP    | HMAP  | LMAP  | HRF   | LRF   | Vanilla |
|--------------|-----------|-------|-------|-------|-------|-------|-------|-------|---------|
| RMSE ($\downarrow$) | **0.076** | 0.089 | 0.107 | 0.119 | 0.090 | 0.091 | 0.096 | 0.093 | 0.115   |
| LPP ($\uparrow$)    | **1.027** | 0.971 | 0.728 | 0.671 | 0.918 | 0.896 | 0.877 | 0.909 | 0.684   |

Table 1: Test-set RMSE and LPP on UCI Energy Efficiency dataset. HF and LF refer to heavy and light fixing with nonzero $c$, HP and LP to heavy and light pruning, HMAP and LMAP to heavy and light fixing with MAP weights, HRF and LRF to heavy and light random (unstructured) fixing.

## 6. Conclusion

Across both experiments, there is a clear relationship between the potency of the symmetry-breaking scheme and the efficacy of the approximate inference scheme. This is an important result as it suggests that parameter symmetry removal could play a key part in achieving accurate and scalable approximate inference in BNNs.

The experimental focus of this work was on fully-connected feed-forward networks under MFVI in just two regression settings. In future work, I aim to refine the schemes for use in more sophisticated architectures, and then to extend the experimental investigation to more diverse networks, applications, and approximate inference schemes. Furthermore, it would be useful to investigate if the proposed symmetry-removal schemes have any adverse effects on the true posterior.

## Acknowledgments

I thank Toby Boyne for his suggestion to compare the method with a partially stochastic BNN *without* a symmetry-breaking structure, as well as Boris Deletic for his suggestions regarding the structure of the manuscript.

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

## Appendix A. Linear-Algebraic View of Structured Partial Stochasticity

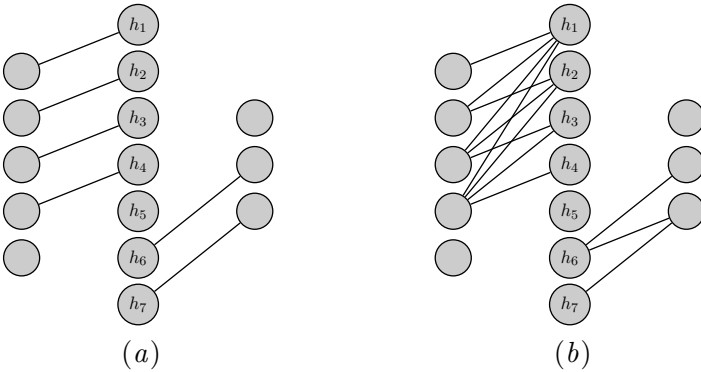

Figure 3: Fixed connections under the respective light and heavy schemes of structured partial stochasticity.

Denoting the activations of the input layer by $\mathbf{x}$, the activations of the hidden layer are given by $\mathbf{h} = \mathbf{x}\mathbf{W} + \mathbf{b}$ (though the diagrams do not show $\mathbf{b}$). Under light structured partial stochasticity, the weights $\mathbf{W}$ take the form:

$$\mathbf{W} = \begin{pmatrix} * & w_{12} & w_{13} & w_{14} & w_{15} & w_{16} & w_{17} \\ w_{21} & * & w_{23} & w_{24} & w_{25} & w_{26} & w_{27} \\ w_{31} & w_{32} & * & w_{34} & w_{35} & w_{36} & w_{37} \\ w_{41} & w_{42} & w_{43} & * & w_{45} & w_{46} & w_{47} \\ w_{51} & w_{52} & w_{53} & w_{54} & w_{55} & w_{56} & w_{57} \end{pmatrix}$$

while under the heavy scheme they take the form

$$\mathbf{W} = \begin{pmatrix} * & * & * & * & w_{15} & w_{16} & w_{17} \\ w_{21} & * & * & * & w_{25} & w_{26} & w_{27} \\ w_{31} & w_{32} & * & * & w_{35} & w_{36} & w_{37} \\ w_{41} & w_{42} & w_{43} & * & w_{45} & w_{46} & w_{47} \\ w_{51} & w_{52} & w_{53} & w_{54} & w_{55} & w_{56} & w_{57} \end{pmatrix}$$

where $*$ represents the presence of a fixed weight (but not necessarily a common value since different fixed weights may take different values in the MAP and nonzero fixing schemes).

## Appendix B. Experimental Details

### B.1. Toy Regression Details

| Hyperparameter | Value |
| --- | --- |
| Architecture dimensions | 1, 50, 50, 1 |
| Activation function | sigmoid |
| Prior | $\mathcal{N}(0, 1)$ |
| Likelihood | $\mathcal{N}\Big(f_{\mathcal{W}}(x), 0.05\Big)$ |
| $c$ | 5.0 |
| MFVI implementation | Reparameterization Trick (Kingma and Welling, 2014) |
| Learning rate | 1e-2 linearly scheduled to 1e-3 |
| Number of training samples | 16 |
| Number of evaluation samples | 300 |
| Epochs | 10,000 |
| Batch size | full batch |
| Optimiser | Adam (default) |
| Datapoint locations | Sampled from $\mathcal{U}(-2.5, -0.75)$ and $\mathcal{U}(0.75, 2.5)$ |
| GP covariance | exponentiated quadratic |
| GP lengthscale | 0.2 |
| Additive Gaussian noise $\sigma$ | 0.05 |

Table 2: Experimental details for the toy regression experiment.

## B.2. UCI Energy Experimental Details

| Hyperparameter | Value |
|---|---|
| Architecture dimensions | 8, 50, 50, 2 |
| Activation function | tanh |
| Prior | $\mathcal{N}(0,1)$ |
| Likelihood | $\mathcal{N}\Big(f_{\mathcal{W}}(x), 0.1\Big)$ |
| $c$ | Optimised jointly with variational parameters |
| MFVI implementation | Reparameterization Trick (Kingma and Welling, 2014) |
| Learning rate | 1e-2 linearly scheduled to 3e-3 |
| Number of training samples | 16 |
| Number of evaluation samples | 1000 |
| Epochs | 10,000 |
| Batch size | 512 |
| Optimiser | Adam (default) |
| Dataset | UCI Energy Efficiency (Tsanas and Xifara, 2012) |
| Dataset train-test split | Random 80/20 (resampled each trial) |
| Seeds | $\{0, 1, 2, 3, 4, 5\}$ |

Table 3: Experimental details for the UCI Energy Efficiency regression experiment.

The reported RMSE was the test-set RMSE of the predictive mean, averaged over repeat trials. The reported LPP was the mean per-test-point LPP, averaged over repeat trials. The LPP probability of a test point $\{x_*, y_*\}$ was estimated by naive Monte Carlo integration of the likelihood of the point under 1000 (approximate) posterior samples:

$$\mathrm{LPP}(x_*, y_*) \approx \log\Big(\frac{1}{1000}\sum_{i=1}^{1000} p\big(y_* | \mathcal{W}^{(i)}, x_*\big)\Big), \quad \mathcal{W}^{(i)} \sim q(\mathcal{W}|\mathcal{D}).$$

This was implemented in a numerically stable manner via:

$$\mathrm{LPP}(x_*, y_*) \approx \mathrm{LogSumExp}\bigg(\Big\{\log\Big(p\big(y_*|\mathcal{W}^{(i)}, x_*\big)\Big)\Big\}_{i=1}^{1000}\bigg) - \log(1000).$$

| | HF | LF | HP | LP | HMAP | LMAP | HRF | LRF | Vanilla |
|---|---|---|---|---|---|---|---|---|---|
| RMSE (↓) | **0.076**±0.011 | 0.089±0.007 | 0.107±0.023 | 0.119±0.007 | 0.090±0.016 | 0.091±0.018 | 0.096±0.021 | 0.093±0.021 | 0.115±0.009 |
| MLPP (↑) | **1.027**±0.046 | 0.971±0.044 | 0.728±0.189 | 0.671±0.080 | 0.918±0.141 | 0.896±0.180 | 0.877±0.217 | 0.909±0.164 | 0.684±0.137 |

Table 4: A copy of Table 1 but with ±1 standard-deviation as error bars.

