# OpenReview forum: "Structured Partial Stochasticity in Bayesian Neural Networks"
_approximateinference.org/AABI/2024/Symposium — AABI 2024_

### Official Review · Reviewer_CmFS · 2024-04-08

**Rating:** 4
**Confidence:** 3

**Review:**

The paper proposes to remove permutation symmetries from DNNs by removing specific edges from the network such that symmetries are impossible. In theory this leads to permutation-symmetry-free networks, whose posteriors are exponentially more efficient. The idea is ingenous!

The results indicate dramatic improvements, which are really impressive.

The paper however suffers from two issues:
- The prunings and fixings should be illustrated in a visualisation. The explanation of the "fixing" was a bit confusing. I did not understand what's going on. Furthermore, it wasn't clear why the paper implies that even after removing edges some symmetries are still possible. It would be good to empirically verify whether permutations still exist or not.
- The results are only from a toy 1D regression and UCI energy on simple MLP. Even for a workshop paper this is way too small, and does not do the main idea justice.

For the above reasons I'm a bit hesitant to vote for acceptance, even at the workshop/poster track.

---

### Official Review · Reviewer_4fdy · 2024-04-17

**Rating:** 6
**Confidence:** 4

**Review:**

This paper studies approaches to remove the permutation invariance of hidden nodes, which removes the abundant modes that correspond to the same network function. Specifically, the authors proposed a heuristic but simple way to force a subset of weights to be 0 (heavy pruning), a learnable constant (heavy fixing, c), and their MAP estimates (heavy fixing, MAP). This improves local approximation methods for BNNs, such as VI and MAP, which often fail to model the repeated modes in the posterior.

Strengths:
1. The proposed method is simple and effective, so it should be applicable to other architectures with minor modifications;
2. The paper is well-written, and the problem is well-motivated.

Weakness:
1. Only fully connected NNs are considered, although the proposed methods can be generalized to other architectures, in my opinion;
2. Experiments don't show error bars.
3. Model hyper-parameters, i.e., scales of Gaussian prior and likelihood, are fixed instead of cross-validated, which might affect the conclusion (A.2.).

---

### Official Review · Reviewer_zHi6 · 2024-04-22
**Well-written paper on a new way of breaking symmetries in Bayesian neural networks**

**Rating:** 7
**Confidence:** 4

**Review:**

This paper describes a new approach for breaking symmetries in the parameters of Bayesian neural networks. The motivation behind the approach is the fact that multiple modes of the posterior distributions of such networks occur due to the symmetries of their parameters, and those modes make posterior inference very difficult. The approach in the paper is structural in the sense that it works on the structure of a network by fixing the weights of certain links. It has an option of fixing a relatively small number of links (in the order of O(n) where n is the number of nodes in a layer), or a large number of links (in the order of O(n^2) where n is again the number of nodes in a layer). The experiments of the paper show that the proposed approach works well for a toy-regression problem and the UCI Energy Efficient regression dataset.

I support the acceptance of the paper. The paper is well-written. I think that the paper can even be a good introduction to the existing (and also the authors') attempts to address the harmful effect of symmetries in Bayesian neural networks. The approach in the paper is simple, and has a potential to be applied to other architectures. Of course, the experiments with more datasets and the handling of other architectures would have made the paper stronger, but I think that they are perhaps something to be done in the full version of the paper, not in this workshop version.

---

### Official Review · Reviewer_zjaP · 2024-04-24
**Removing symmetry and removing posteior modes**

**Rating:** 6
**Confidence:** 4

**Review:**

This paper explores how parameter symmetry in BNNs can result in multi-modal posteriors, making inference more challenging. Literature has shown that by removing parameter symmetries we can simplify the posterior space with hopes to make inference more feasible. Two methods are proposed; a light fixing/pruning and heavy fixing approach, where the heavy fixing approach will account for practical scenarios where light pruning may not necessarily remove certain modes.

Experimental results shows an improved predictive distribution for synthetic regression task in terms of fitting to the data and uncertainty quantification, Following experimentation demonstrates that the pruning method yields improved predictive performance in terms of RMSE and posterior predictive likelihood. A limitation for this section here is simply due to the number of data sets explored. The results do indicate promise that the removal of weight symmetries can improve inference, though the empirical evidence is not strong enough to draw definitive conclusions. The discussion section highlights how future work is planned to look at more of these settings which is great to see. Beyond this I believe there is more avenues that are worth exploring following this paper. In the experimentation the Heavy fixing outperformed the Heavy MAP methods, which I find counter-intuitive or at least a little surprising, especially given that in the introduction the authors state that when finding a MAP estimate that the parameter symmetries present much less of an issue. This to me suggests some form of further inductive bias being induced due to the weight sharing nature imposed on this type of MLP? I suspect that their might be some intricate details within the MAP learning scheme for weight fixing when performing inference jointly that might be able to overcome this issue seen.

Overall I think there is value from this paper and would recommend it marginally above the acceptance threshold. The concept is quite simple (which I mention as positive), and is work that is building off the growing literature exploring ways to simplify the posterior space to improve inference. With further development within this direction I expect to see some interesting findings (either positive or negative). One minor suggestion I pass to the authors is within some of the language presented, in that there are certain statements which I think may be too strong given the theory and evidence presented. Statements in the discussion such as "parameter symmetry removal could be the key to scalable and accurate approximate inference in BNNs." Whilst I do believe there is benefit in exploring this work even further, I am doubtful that removing parameter symmetry could be the "key" so to speak. It might help, but for the performing inference over such high dimensions I expect that more work will be needed to further make Bayesian inference for NNs more feasible and performant. For example, long-tailed posteriors may make exploring the typical set difficult, even if unimodal, or the resulting posterior might be heavily skewed, or strong covariance may make inference difficult, etc. I think there is a lot of challenges with inference in BNNs, and that exploring parameter symmetry may indeed help us, but I expect there is a lot of work ahead to deal with all the issues that might arise. Also in the introduction it is stated that by removing symmetry that the posterior will be unimodal. I dont believe this in entirely correct, in that depending on the data/prior/likelihood, that the posterior may still be multi-modal, rather that additional modes due to symmetry will no longer be present. The abstract also states the posterior is "drastically simplified". Im not necessarily convinced on the "drastically" component here, just as there is no explicit evidence to show exactly how much posterior is being simplified through the fixing operation. This also ties into the inference scheme being used. Perhaps symmetry might be less of an issue with different inference approaches?

---

### Official Review · Reviewer_LcYb · 2024-04-24
**Evaluation can be more convincing**

**Rating:** 4
**Confidence:** 3

**Review:**

The paper deals with the problem of treating only a subset of the weights of a neural network as stochastic while treating the remaining one as deterministic. The main contribution of the paper is a new way to select these weights, which they referred to as ''heavy pruning'', which should lead to an easier posterior, i.e. a posterior without redundant modes created by the permutation symmetries. This should make it easier to approximate the posterior distribution. Instead of pruning, which corresponds to setting the value of those weights to 0, they also consider the case of fixing their values to a specific constant $c$ and they explore also the case where $c$ is the value of the MAP estimate for those specific weights.


**Strengths**
- The heavy pruning idea seems novel in the context of sub-network approximate inference and it is an interesting idea for removing permutation symmetry modes in the posterior.
- From the two experiments presented in the paper it seems that the proposed approach improves over vanilla mean-field VI using all the weights, but evaluation can be improved (see below).

**Things that can improve the paper**
There are some aspects of the paper that are not fully convincing and I think that addressing these points will improve the message of the paper:
- I feel the selection of $c$ is problematic and not well understood. In the regression experiment, I feel like the final results heavily depend on the value of $c$. So how can one select that value? And also how do you decide if you use $+c$ or $-c$? In the MNIST example you optimise the value of $c$ instead, what value did you get for $c$ then? how does it looks like for the regression case if you optimize it as done in the MNIST example?
- In the regression experiment I would be interested to see also just the MAP results. Is the MAP fitting the data properly? Then do you have any intuition as to why fixing some of the weights to the MAP values and running VI gets worse results? Are you running VI still with an N(0,1) prior even though maybe the MAP value of those weights that you are treating stochastically was way off? Meaning that you already know that by using such prior is really unlikely to get a mean value of the weights that actually makes sense. Also since the considered network is small it would be interesting to see how getting samples from the full posterior using HMC looks like.
- In the regression example, why are you not also comparing with the light pruning strategy?
- The method is only tested on small MLPs therefore we are not sure about the effect of this pruning strategy when considering bigger networks.
- For Table 1 you mention in the text that you average over 6 repeat trials but you are not reporting any standard deviation on the final results.

---

### Official Review · Reviewer_JgLZ · 2024-04-25
**The paper proposes a new way of prunning MLPs that could break the permutation symmetry problem of them**

**Rating:** 7
**Confidence:** 3

**Review:**

Strength:
+ First, the authors show how they can break symmetries in a neural network by merely removing a few connections.
+ The authors propose an interesting hypothesis explaining why partial stochastic networks may outperform full stochastic counterparts: due to their potentially lower degree of symmetries.
+ The proposed restructuring of MLPs is tested in two setups, and their effectiveness is validated in simple setups.
- While the explanation covers how an MLP can be designed to eliminate permutation symmetry, it does not extend to other types of layers.
- The experiments are done on relatively small MLPs, raising questions about the method's generalization to larger networks.

Clarity: The paper is well-written. It would be helpful if the authors include a schematic illustrating how pruning is performed in the MLP.

Originality: The proposed ideas appear to be novel.

Significance of this work: The structuring algorithm is proposed within an MLP setting, potentially limiting its applicability. Furthermore, the experiments only test small MLPs. However, the paper presents interesting ideas that merit acceptance.

Questions:
- Is any test done with larger MLPs?
- Can authors comment on how to extend the pruning beyond MLPs?

---

### Meta-Review · Area_Chair_Uy3Q · 2024-05-26

**Recommendation:** Accept (Poster)
**Confidence:** 4

**Metareview:**

The reviewers have shared detailed feedback and highlighted several strengths and weaknesses of the paper. Two reviewers recommended acceptance, two reviewers recommended rejection, and one reviewer considered the paper to be marginally above the acceptance threshold. Having read the reviews, I believe that the paper makes an interesting contribution and should be presented at the symposium. However, the reviewers have highlighted several issues that I strongly recommend authors address for the camera-ready version of the manuscript. I recommend acceptance.

---

### Decision · Program_Chairs · 2024-05-27

Accept